# Combination Modality Using Quercetin to Enhance the Efficacy of Docetaxel in Prostate Cancer Cells

**DOI:** 10.3390/cancers15030902

**Published:** 2023-01-31

**Authors:** Satish Sharma, Katherine Cwiklinski, Supriya D. Mahajan, Stanley A. Schwartz, Ravikumar Aalinkeel

**Affiliations:** 1Department of Medicine, Clinical and Translational Research Center, Division of Allergy, Immunology and Rheumatology, Jacobs School of Medicine and Biomedical Sciences, University at Buffalo, Buffalo, NY 14203, USA; 2Department of Urology, Jacobs School of Medicine and Biomedical Sciences, University at Buffalo, Buffalo, NY 14203, USA

**Keywords:** combination drug treatment, quercetin, docetaxel, flavonoids, apoptosis

## Abstract

**Simple Summary:**

Currently, castration-resistant prostate cancer has limited therapeutic options and is incurable after it sets in. We undertook this research to see if combination therapy with plant derived natural products are of help in overcoming the limitations of the current standard of care drug regimen. Our results indicate that a combination treatment with the natural flavonoid quercetin sensitizes the cancer cells to lower doses of the standard of care drug docetaxel, thereby reducing drug induced toxicity, and shows additive and synergistic effects with docetaxel for efficient cancer cell killing. We have identified optimum combination doses and treatment approaches for this purpose. The results of the present study provide the research community a novel therapeutic modality to enhance the efficacy of Doc in a nontoxic manner.

**Abstract:**

The standard of care chemotherapy drug presently used to treat castration-resistant prostate cancer (CRPC), docetaxel (Doc), also develops chemoresistance, thereby reducing its clinical utility. Since resistance to chemotherapy drugs can be overcome by co-treatment with plant-based bio-active compounds we undertook the present study to evaluate if quercetin (Que), a flavonoid present in plants such as onions, apples, olives, and grapes can enhance the efficacy of Doc. We studied the separate and combined effects of Que and Doc at different doses and different combination approaches in two different prostate cancer cell lines, DU-145 (moderately aggressive) and PC-3 (very aggressive), and assessed the effects of these combinations on viability, proliferation, and apoptosis. Monotherapy with these drugs showed dose-dependent cytotoxicity; however, only Doc monotherapy showed a statistically significant difference in IC_50_ levels (IC_50_ = 4.05 ± 0.52 nM for PC-3 and IC_50_ = 2.26 ± 0.22 nM for DU-145). In combination treatment, we used three different treatment approaches (TAP). The concentrations and range analyzed were chosen based on the approximate cytotoxicity of 30–50% when the drugs were used individually. Our observations indicate that the most beneficial effect of the Que and Doc combination was obtained with the TAP-2 approach, which is pre-treatment with all doses of Que for 24 h followed by low doses of Doc for another 24 h. Using this approach, we observed synergism at low concentrations of Doc (0.5 and 1.0 nM) and all concentrations of Que. An additive effect was observed at moderate and high concentrations of Doc (1.5, 2.0, and 2.5 nM) and all concentrations of Que in both cell lines. The TAP-2 strategy was also helpful in overcoming Doc resistance in resistant CaP cells. In summary, Que improved the therapeutic effect of Doc in CRPC, and it is proposed that this improvement is mediated through multiple mechanisms. This study provides a novel therapeutic modality for an effective combination using Doc and Que to enhance the efficacy of Doc in an innocuous manner for Doc resistance and CRPC treatment.

## 1. Introduction

According to the National Cancer Institute-Surveillance, Epidemiology, and End Results (NCI-SEER) database, prostate cancer (CaP) is the most diagnosed noncutaneous cancer in American men and is the second leading cause of cancer-related deaths in men in North America [1]. It is estimated that in 2022, ~268,490 new cases of CaP will be diagnosed and nearly 34,500 men will die of this disease [2]. For most men diagnosed with localized CaP, the first line of curative therapy is surgery or radiation; however, the major public health burden comes from the metastatic stage of CaP, for which there are currently limited remedial options with a 5-year survival of less than 30% [3]. The major therapy for advanced and metastatic CaP is androgen deprivation therapy (ADT), which has a variable but a short time (3 ± 0.34 years) for tumor progression [4]. When CaP progresses after ADT, the disease becomes hormone-refractory and often metastasizes from the prostate to the bone. This stage is referred to as castration-resistant prostate cancer (CRPC) and there are no effective treatments for CRPC. Most CaP-related deaths are attributed to metastatic dissemination, predominantly to the bone and lungs [5]. Although these cancers become refractory to ADT, CaPs retain a functional androgen receptor (AR) [6].

Docetaxel (Doc), a trailblazing new drug for treating CRPC, was approved by the US Food and Drug Administration (FDA) in 2004 [7,8]. At present, this drug is the standard treatment for CRPC and remains a mainstay in the current drug development for CRPC [9]. Doc exerts its cytotoxic activity through the stabilization of microtubules, filamentous polymers composed of α- and β-tubulin heterodimers critical for cell division [10]. Doc chemotherapy in CRPC shows prolonged median overall survival and exhibits palliative benefits, but chemoresistance to this drug does set in [7,8]. Furthermore, ~50% of patients with CRPC do not respond to Doc therapy, presenting a significant clinical problem [7,8]. Moreover, initial responders to Doc treatment ultimately develop Doc resistance. While the exact mechanism for Doc resistance is yet to be elucidated, de novo and acquired resistance to Doc chemotherapy are the main limitations to its efficacy. Therefore, there is an urgent need to develop new and effective drug combinations or modulators of Doc therapy to overcome Doc resistance and lower its doses in CRPC to avoid the development of Doc resistance. Even though several therapies for the reversal of Doc resistance have been reported, their progression to clinical reality is far away, mainly due to their adverse side effects [11,12].

Many studies have focused on ways to complement or synergize cancer therapy using natural products as an adjunct treatment or with plant-derived bioactive compounds, which can modulate the treatment efficacy of chemotherapeutic drugs [13]. One such natural compound is quercetin (Que), a flavonoid group of polyphenols and a major bioflavonoid in the human diet that has chemopreventive, antiproliferative, cell death and other bioactivities [14,15]. This has led to a plethora of studies on this compound, revealing numerous pathways that either independently or synergistically act to prevent or treat cancer [16,17]. Such studies have also revealed that Que exerts a biphasic dose response on cells depending on its concentration. The chemopreventive effects of Que were observed in the 1–40 μM range and were likely mediated by Que’s antioxidant properties. The strong antioxidant activity of Que maintains a good oxidative balance and plays a substantial part in the chemoprevention. High amounts of reactive oxygen species (ROS), which cause oxidative stress, promote cell proliferation, survival, and metabolic adaption to the tumor microenvironment by over-activating signal transduction pathways involved in tumor growth. By directly restricting the amount of ROS generated, Que is able to suppress most of these tumorigenic processes and this activity of Que is primarily mediated through its effect on glutathione (GSH). The pro-oxidant effects of Que were present in the 40–100 μM range. However, at higher concentrations, many novel pathways in addition to reactive oxygen species (ROS) contribute to its biological effects [14]. Mechanistic studies on biological effects have demonstrated that the anti-proliferative effect of Que is exerted by producing arrest in the G1 phase of the cell cycle [18] or through interaction with cell cycle-regulated proteins, such as cyclin D1 and CDK-4 [19]. This mechanism of action and at the doses studied clearly indicates a chemotherapeutic role for Que. Our studies have shown that Que exerts significant anti-viral and anti-tumor effects, possibly by modulating the production of Th-1 and Th-2-derived cytokines which is an immune boosting action suggesting a chemopreventive role for Que. [20,21]. Specifically, in trying to understand the role of Que in CaP chemotherapy, we demonstrated that it inhibits the colony-forming ability and expression of genes related to cell cycle regulation and tumorigenesis [21]. A subsequent study by us to determine the mechanism underlying Que’s actions as a chemotherapeutic agent on CaP cells and to evaluate the potential pharmacological use of this flavonoid for CaP identified heat shock protein 90 (HSP-90) as an important molecular target of Que [22]. Our study further demonstrated that along with HSP-90, the levels of human epidermal growth factor receptor-2 (HER-2), and Insulin-like growth factor binding protein-2 (IGFBP-2) were also reduced as the concentration of Que increased in CaP cells [22]. Conversely, we also found that Que treatment in healthy prostate epithelial cells did not show this effect [22]. Several studies have reported other mechanisms of action of Que as a chemotherapeutic agent, including inhibitory effects on multidrug resistance-associated protein, p-glycoprotein, and catechol-O-methyltransferase, leading to a synergistically enhanced inhibition of xenograft prostate tumor growth in severe combined immunodeficiency (SCID) mice [23,24,25,26]. Additional chemotherapeutic mechanisms of action of Que on CaP cells have reported the activity to be mediated via induction of apoptosis, inhibition of proliferation and insulin-like growth factor (IGF)-1 pathways [27,28,29]. Que’s antiapoptotic property has also been seen in other cell cancers, making it a worthwhile compound for use in CaP chemotherapy [30,31]. 

Despite evidence supporting the usefulness of Que in cancer therapy, except for a few studies, none have examined the impact of Que on sensitizing cancer cells to chemotherapy [24,32], and only one study has shown the effect of Que on reversing Doc resistance [24,33]. However, even in these studies, the range of concentrations where Que and Doc can act synergistically has not been systemically documented. In the present study, we conducted extensive Que and Doc drug combination studies and assessed the combination effects on cell viability, proliferation, and apoptosis assays to select optimal treatment approaches for DU-145 (moderately aggressive) and PC-3 cells (very aggressive) CaP cells. Based on the results of these combination studies, we further verified whether the most beneficial treatment approach (TAP) can be used to reverse Doc resistance in Doc-resistant CaP cells generated in our laboratory. Our observations indicate that the most beneficial synergistic effect of Que and Doc combination was obtained by pre-treatment with all doses of Que for 24 h followed by low doses of Doc for another 24 h. This combination TAP strategy was also helpful in overcoming Doc resistance in resistant CaP cells. We anticipate that the optimal combination and concentrations of both drugs determined in this study will be used to propose not only a new therapeutic regimen but also promising drug combination delivery systems with Doc in CRPC treatment for potentially reversing the Doc resistance phenotype in CRPC. 

## 2. Materials and Methods

### 2.1. Cells and Culture Conditions

PC-3 and DU-145 cells were obtained from the American Type Culture Collection (Manassas, VA, USA). Both the cell lines were at passage 25 when the experiments were performed. These cells differ in their aggressive behavior significantly, and we have documented this in an earlier publication [34]. DU-145 was originally isolated from brain metastases and PC-3 was isolated from lumbar vertebrae. Both these cell lines are androgen-independent, and they were grown at 37 °C in a humidified atmosphere of 95% air and 5% CO_2_ in RPMI-1640 medium supplemented with non-essential amino acids, l-glutamine, a twofold vitamin solution (Life Technologies, Grand Island, NY, USA), sodium pyruvate, Earle’s balanced salt solution, 10% fetal bovine serum, and penicillin and streptomycin (Flow Labs, Rockville, MD, USA). The cells were harvested by trypsinization. The cell lines used in this study were examined by the eMycoPlus Mycoplasma PCR Detection Kit (BulldogBio, Portsmouth, NH, USA) to confirm that they did not have any mycoplasma infection.

### 2.2. Drug Combination Studies

Evaluation of the combined effects of Doc with Que on cell viability and apoptosis was performed as described individually for each of these assays below; however, the drugs were combined (in a non-constant ratio), rather than used separately. Briefly, the selected concentrations of Doc were mixed with the selected concentrations of Que. Concentrations of the drugs’ solvents were corrected in all wells (including control wells) to a constant level corresponding to the highest used concentration of a particular solvent. The nature of the interaction between Que and Doc was assessed using different treatment approaches: Treatment Approach-1(TAP:1), simultaneous treatment with both drugs for 24 h; and/or (TAP:2), pre-treatment with Que for 24 h, followed by treatment with Doc alone for another 24 h; and/or (TAP:3), pre-treatment with Doc for 24 h, followed by co-treatment with Doc and Que for another 24 h. The cytotoxic, and pro-apoptotic effects of the Doc-Que combination against CaP cells over a range of concentrations were compared with those obtained for the individual drugs. A measure of the synergy between the two drugs, referred to as the combination index (CI), was calculated using CompuSyn software (ComboSyn, Inc., Version 1.0, Paramus, NJ, USA) developed based on the median effect mathematical algorithm [35]. The following assumptions were made: a drug combination was synergistic if its CI value was below 0.9; the combination was additive when the CI was between 0.9 and 1.1; and the combination was antagonistic for CI values above 1.1 [36].

### 2.3. Cell Proliferation Assay

DU-145 and PC-3 cells were seeded in 96-well plates at a density of 1 × 10^4^ cells per well, allowed to adhere overnight and treated with either Doc or Que. The cells were washed and re-fed fresh medium alone or with a medium containing different treatment combinations. At the end of incubation, viability was determined by the 3-(4,5-Dimethylthiazol-2-Yl)-2,5-Diphenyltetrazolium Bromide (MTT) assay 25 using a microtiter plate reader (Bio-Tek Instruments, Inc., Winooski, VT, UK) at 570 nm. Cell proliferation/inhibition was calculated as the percentage of proliferation/inhibition: [1 − (A/B)] × 100, where A is the absorbance of treated cells, and B is the absorbance of untreated control cells expressed as a percentage of control cells. IC_50_ (half maximal inhibitory concentration) values were determined for each drug at each treatment time. The samples were prepared in triplicate.

### 2.4. Docetaxel Resistant Cell Line Development

Since there is no consensus in the literature on the establishment of Doc-resistance, we employed a strategy that has been found effective in our laboratory which consists of sequential exposure of CaP cells to increasing concentrations of Doc to obtain resistance to Doc in DU-145 and PC-3 cells. Briefly, both CaP cell lines were incubated with an initial concentration of Doc (0.1 nM) Doc for 24 h. After 24 h, the dead cells were cleared, and the surviving cells were collected and exposed to the next higher concentration of 0.2 nM. The cycle was repeated for 0.5, 1.0, 2.0, 5.0 10.0, and 25 nM of Doc. The cells that were resistant to 25 nM of Doc were collected and labelled as DU-145/R and PC-3/R, respectively, and were used in the Doc-resistant reversal studies reported in this manuscript. During the step-by-step elevation of Doc concentration, the conditions of cells were under close monitoring. If many dead cells appeared or many cells cloned very slowly, the Doc concentration increase was delayed. Once cells were freely dividing in the medium with 25 nm Doc, they were considered docetaxel-resistant, labeled as DU-145/R and PC3/R, respectively, and were used for the studies related to Doc resistance reversal.

### 2.5. Cell Invasion Assay

The effect of combination therapy treatment on the invasive activity of CaP cells in vitro was tested using a Quantitative Cell Invasion Assay kit (Chemicon International, Inc., Temecula, CA, USA). The invasion assay was performed using 24-well, Transwell tissue-culture plates. The insert contains an 8-μm porous polycarbonate membrane pre-coated with basement membrane proteins derived from the Engelbeth Holm–Swarm mouse tumor. Details of the methodology have been described previously [34]. 

### 2.6. Wound Healing Assay

A wound healing assay was used to measure cell migration capability. Both Du-145/R and PC-R were plated in 6-well plates. Afterwards, a sterile 200-μL pipette tip was used to scrape a vertical wound. The cell debris was removed by washing with PBS twice and the cells were treated with serum free RPMI 1640 medium containing the following: vehicle control (DMSO), 20 μM Que, 1.0 nM Doc, and 20 μM Que + 1.5 nM Doc. The assays were performed in triplicate, and at the end of the incubation period, images were obtained with an inverted microscope at base line and on termination (48 h). The images and the area of wound healing were calculated by Image J software (Scion Corp., Frederick, MD, USA).

### 2.7. Clonogenic Assays

DU-145, PC-3, DU-145/R, and PC-3/R cells were cultured in a complete RPMI medium and treated for 48 h with either vehicle, Doc (1.5 nM), Que 20 μM, or a combination of Doc and Que (1.5 nM Doc + 20 μM Que). Following a 48-h incubation with the different drug combinations, the cells were trypsinized to be plated for clonogenic assays. Clonogenic assays were performed on the collected cells per published methods [37]. Briefly, ~400 cells obtained after trypsinization with each drug were seeded in 60-mm tissue culture dishes in 5 mL of the growth medium, and the dishes were returned to the incubator for up to 8 days without a change in medium. When the colonies were distinct and well-defined, the dishes were washed with PBS, fixed in methanol, and stained with Giemsa (Sigma-Aldrich) before drying. The number of colonies per dish was counted using a computer-based Gene Tool software Version 2.11a (Syngene, Frederick, MD, USA). The assays were repeated three times for each cell line. 

### 2.8. Apoptosis Assay

For the apoptosis assay, cells were seeded at 18 × 10^3^ cells/250 μL medium/0.95 cm^2^ growth area in 48-well plates, grown overnight, and treated with Que or Doc for 24 h and/or 48 h. The concentrations of the drug solvents were corrected in all wells (including the control wells) to a constant level corresponding to the highest used concentration of a particular solvent. After the treatments, the assay was performed using the FITC Annexin V/Dead Cell Apoptosis Kit with FITC annexin V and PI (Invitrogen™, Grand Island, NY, USA) according to the manufacturer’s protocol. The cells were counted with a fluorescence microscope Eclipse Ti (Nikon Instruments Inc., Melville, NY, USA). For each well, the ratio of apoptotic cells to the total number of cells in three different fields was calculated. The samples were prepared in triplicate.

### 2.9. Statistical Analysis

All the experiments were repeated at least three times in triplicate. Values are expressed as the mean ± SD. The significance of the difference between the control and each experimental test is analyzed by an unpaired *t*-test (GraphPad, Prism 9.0, Boston, MA, USA) and a value of *p* < 0.05 was considered statistically significant. 

## 3. Results

### 3.1. Monotherapy of DU-145 (Moderately Aggressive) and PC-3 (Very Aggressive) CaP Cells with Docetaxel and Quercetin

To determine whether Que can modify the cytotoxic effect of Doc on CaP cells, we first evaluated the effect of Doc and Que independently on the viability of two aggressive CaP cell lines. When DU-145 (moderately aggressive) and PC-3 cells (very aggressive) were treated with Doc in the nM range for 24 h, the viability of both cells was dose-dependently inhibited by Doc (both *p* < 0.001 compared with untreated cells; Figure 1A). However, the calculated IC_50_ values (IC_50_ = 4.05 ± 0.52 nM for PC-3 and IC_50_ = 2.26 ± 0.22 nM for DU-145) were statistically different (*p* < 0.001; Figure 1D). On similar lines when treatment with Que was assessed for its cytotoxicity on both the cell lines, we observed a cytotoxic effect in a dose- and time-dependent manner (Figure 1B,C), but there were no significant differences between IC_50_ values obtained for both cell lines during 24-h treatment (IC_50_ = 32.59 ± 2.55 μM and 37.85 ± 1.54 μM for PC-3 and DU-145, respectively; Figure 1D) or 48-h treatment (IC_50_ = 32.52 ± 7.77 μM and 35.09 ± 6.59 μM for PC-3 and DU-145, respectively; Figure 1E).

### 3.2. The Effect of Quercetin and Docetaxel Combination on DU-145 (Moderately Aggressive) and PC-3 Cells (Overly Aggressive) CaP Cells

After confirming that both Doc and Que are cytotoxic, we performed experiments employing Que-Doc combinations to assess the degree of interactions between the two agents and select the most effective combination doses for further investigation in Doc-resistant CaP cells. The concentrations and range analyzed were chosen based on the approximate cytotoxicity of 30–50% when the drugs were used individually. At the end of each combination dose and TAP, we calculated the combination index (CI) using CompuSyn software (ComboSyn, Inc.). The calculated r values by the CompuSyn software were greater than 0.95, as described elsewhere in cell culture experiments [38,39]. The results are shown as a heat map in Figure 2. For these experiments, as explained in Section 2, we used three different treatment approaches (TAP). TAP-1 involved simultaneous treatment with different doses of both Que and Doc for 24 h, TAP-2 was a pre-treatment with Que for 24 h followed by treatment with Doc for another 24 h, and TAP-3 consisted of pre-treatment with Doc for 24 h followed by simultaneous treatment with Que and Doc for another 24 h. When TAP-1 was used in DU-145 cells (Figure 2A), our results show synergism with a combination of low concentrations of Doc (0.5 and 1.0 nM) and low concentrations of Que (2 and 5 μM). Furthermore, we observed an additive effect at low concentrations of Doc (0.5, 1.0 nM) and high concentrations of Que (10 and 20 μM) and an additive effect at a medium concentration of Doc (1.5 nM) and all concentrations of Que. However, antagonism was the only observation at high concentrations of Doc (2 and 2.5 nM) and all concentrations of Que. When the TAP-2 method was employed in DU-145 cells, we observed synergism at low concentrations of Doc (0.5 and 1.0 nM) and all concentrations of Que and an additive effect at a moderate and high concentration of Doc (1.5, 2.0, and 2.5 nM) and all concentrations of Que. However, when the TAP-3 strategy was employed, we observed strong antagonism (CI >> 1.1) at all combinations of both the drug combinations used. Similar to DU-145 cells, when PC-3 cells were treated with Que and Doc simultaneously, (TAP-1, Figure 2B), we observed synergy at low and moderate doses of Doc (0.5 and 1.0, and 1.5 nM) and at all doses of Que. However, at higher concentrations of Doc (2 and 2.5 nM) and at all concentrations of Que, we observed antagonism. Alternatively, when we employed TAP-2 in PC-3 cells, we observed synergy at low concentrations of Doc and all concentrations of Que and additive effects at high concentrations of Doc and all concentrations of Que, as observed in DU-145 cells. However, when PC-3 cells were pretreated with Doc for 24 h, followed by co-treatment with Que and Doc for another 24 h (TAP-3), we observed antagonism at all combination concentrations of both drugs studied, as in DU-145 cells. Based on the results of the combination studies, we conclude that TAP-1 is beneficial at low and moderate concentrations of Doc and all concentrations of Que and TAP-3 is not at all beneficial at any concentrations of both drugs since the main observation was antagonism. However, since TPA-2 shows better synergy or additive effects at all concentrations of both drugs studied, we conclude that TAP-2 at low concentrations of Doc is the most beneficial TAP for both the cell lines. 

### 3.3. The Generation of Docetaxel Resistant Prostate Cancer Cell Lines

The results of the studies with our Que-Doc combination and TAP-2 further led us to examine whether this approach, in addition to having a Doc sensitization effect can also have a Doc-resistant (Doc-R) reversal effect in Doc-R CaP cells. For this purpose, we first established Doc-R phenotypes, DU-145/R and PC-3/R, in our laboratory, as described in Section 2. Following the establishment of the Doc-R phenotypes DU-145/R and PC-3/R, we validated the true resistance of these cells to Doc treatment in a 24-h treatment assay. Our results show that the proliferation of parental CaP cells (Doc-naïve cells) was inhibited in a dose-dependent manner (Figure 3A,B). In these studies, even at a low concentration of Doc (1.0 nM), we observed a significant growth inhibition in both Doc naïve cells. In comparison, at the same concentration, the cell proliferation was not affected in either Doc-R cell lines, which confirms the Doc-R nature of the CaP cells that we generated. In addition, we also observed significant growth inhibition in both the parental and naïve cells of both cell lines in a dose dependent manner to different concentrations of Que (Figure 3D,E). In all studies, DMSO, which was used as a solvent, did not affect the cell viability.

### 3.4. TAP-2 Doc-Que Combination Effect on Doc Resistance Reversal in Doc-Resistant CaP Cells DU-145/R and PC-3/R

After establishing the Doc-R CaP cells and the most beneficial approach for combination tested in our earlier experiment, we further analyzed whether the most beneficial TAP treatment for effective CaP cell sensitization would influence the reversal of Doc-R as well. To understand this, we analyzed the effects of the drugs independently and in combination using cell proliferation, cell migration, cell invasion, and colony formation assays in both DU-145/R and PC-/R cells. Our results on cell proliferation show that a high concentration of Que (20 μM) showed a significant reduction in the cell proliferation rate at both 24 h and 48 h (Figure 3C,F). However, Doc alone at a concentration of 1.0 nM failed to induce any anti-proliferative effect on DU-145/R or PC-3/R cells. In addition, the results of these studies show that the antiproliferative rate of combination therapy based on the most beneficial combination approach, TAP-2 (pre-treatment with 20 μM Que + 1.0 nM Doc), was much more effective than that of Que (20 μM) alone in both cell lines. 

To assess TAP-2 reversal on the migration potential of CaP cells, a wound-healing assay was performed with 20 μM Que and 1.0 nM Doc. The results of this study show that the wound healing difference between DU-145/R and PC-3/R cells was not significant. In the Doc treated group, the wounds were only slightly open and the difference between the Doc group and the control group was insignificant in both the cell lines (Figure 4A,B). However, the wound was partially open in the Que group and almost did not heal in the combination group for both the cell lines. To further understand the effect of our combination modality using TAP-2 on the reversal invasive potential of the Doc resistant CaP cell lines, a cell invasion assay was performed. The results of this study show that similar to the cell migration assay, Doc (1.0 nM) by itself did not inhibit cell invasion compared with the vehicle control in both the resistant cell lines (Figure 5A,B). However, we did observe the number of cells invading through the Matrigel to be significantly inhibited by Que (20 μM) alone compared with the vehicle control group, and the number of cells invading through the Matrigel was maximumly inhibited in the combination group among the four groups that we studied. To examine the effect of this treatment on colony formation in PC-3/R and DU-145/R, both the cell lines were pre-treated with Que for 24 h followed by Doc for an additional 24 h, and an equal number of surviving cells were seeded for colony formation. Our results show that with the treatment of 1.0 nM Doc, the colony formation ability of DU-145/R or PC-3/R cells was not inhibited at any significant level compared with that of the control group (Figure 6A,B). However, there was a significant reduction in the number of colonies in the combination therapy group (20 μM Que + 1.0 nM Doc, *p* < 0.05). Taken together, these results indicate that pretreatment with Que for 24 h followed by Doc for an additional 24 h has a Doc-resistance reversal effect, as demonstrated by a significant decrease in the proliferation, migration, invasion, and colony formation of Doc-resistant CaP cells.

### 3.5. Docetaxel-Resistance Reversal Effect of TAP-2 on Apoptosis

We further investigated by flow cytometry whether the TAP-2 approach would be beneficial in the induction of apoptosis in DU-145/R and PC-3/R. The results of this experiment (Figure 7) showed no apoptosis-inducing effect in either cell line when Doc was used alone in the 1.0 nM Doc groups (Doc group vs control group: *p* > 0.05). In the same experiment, we also observed the percentage of DU-145/R and PC-3/R cells with positive Annexin V-positive staining (total apoptotic cells) increased with the treatment with 20 μM Que (Figure 7A,B). However, there was a significant increase in the apoptosis in response to TAP-2 (20 μM Que + 1.0 nM Doc) compared with the quercetin (20 μM) only group (*p* < 0.001). We conclude that the TAP-2 approach using pretreatment with Que showed a Doc-resistance reversal effect but also induced apoptosis in Doc-R CaP cells.

## 4. Discussion

Prostate cancer is the second most prevalent cancer in men worldwide, accounting for ~1.4 million cases globally per year [40]. However, regarding mortality, CaP, is in the fifth position, accounting for about 6.8% of total cancer mortality globally (totaling ~3, 74,000/yr) [41]. Metastatic CaP has a negative prognosis and is rarely curable. Management strategies for symptomatic metastatic CaP typically involve therapy directed at the relief of symptoms (e.g., palliation of pain) and attempts to slow the further progression of the disease using hormone therapy, which mainly involves ADT [42]. ADT may be either surgical castration (orchiectomy) or chemical castration. Agents used for chemical castration include luteinizing hormone-releasing hormone (LHRH) analogs or antagonists, antiandrogens, and other androgen suppressants [42]. CaPs are classified as CRPC when they no longer respond to hormone therapy and usually have metastatic lesions. Most of the deaths associated with CaP fall into this category. In the setting of CRPC disease, Doc chemotherapy provides modest symptomatic and survival benefits [12]. However, the development of chemoresistance to Doc is observed in most patients and limits its clinical success. Additionally, some severe side effects are associated with Doc treatment, including the suppression of bone marrow function, leading to immune dysfunction and anemia as major issues [43].

Natural dietary phytochemicals have recently attracted attention because they can be used alongside conventional chemotherapy to improve clinical response [44]. In terms of improved clinical response, a natural flavonoid called Taxifolin (dihydroquercetin) was recently shown to have improved clinical response in ischemic insults by providing protection to enzymatic antioxidant systems [45]. Further, the delivery of Taxifolin as a selenium–taxifolin nanocomplex showed good cytoprotective efficiency in the normal cortical cells during ischemia/reoxygenation and the action of exogenous H_2_O_2_ [46]. Numerous dietary substances, including capsaicin [38], curcumin [38,47], thymoquinone [39], epigallocatechin gallate [44], methoxyphenyl chalcone [48], and delphinidin [49], can make CaP cells more sensitive to Doc when used together. Que is a natural flavonoid that is found in various fruits and vegetables, including red wine, green tea, onions, and apples, and has multiple beneficial attributes, such as antioxidant, anti-inflammatory, and anti-cancer activities both in vitro and in vivo [14]. Our earlier studies showed Que’s ability to inhibit the proliferation of human CaP cells via the HSP-90-mediated pathway [22]. Additionally, Que has no such effect in normal prostate cells. Mounting evidence indicates that Que makes some chemotherapeutic treatments more effective against cancer cell lines that are multidrug-resistant [25,26]. Ferry and colleagues investigated the pharmacokinetic effects of intravenously injected Que in patients with cancer at doses of 60–2000 mg/m^2^, and a safe dose of 945 mg/m^2^ was identified [50]. At higher doses, Que was toxic, and vomiting, high blood pressure, nephrotoxicity, as well as decreased serum potassium concentrations were documented. Despite these studies, there are no available data systematically studying the combination of Doc and Que at a TAP that is safe and efficacious in CRPC therapy. In this study, we determine the most beneficial combination doses of Doc and Que and demonstrate that co-treatment of the CaP cells with selected concentrations of Que and Doc involves a synergism in the interaction of these two drugs, leading to sensitization, reduced viability, and increased apoptotic rate of the CaP cells, thereby lowering the required concentrations of Doc.

The results of our study on the dose-dependent cytotoxicity of Doc on CaP cells generally mirror the results of other studies on Doc toxicity in CaP cells [51,52], and overall, are consistent with the toxicity reported on the use of Doc in other cancers [53]. This observation is easily explained by the fact that Doc, like other taxanes, works by attaching to microtubules and altering cell signaling and cytoskeleton development in CaP and other cancers [54]. The antiapoptotic protein Bcl2 is also known to be phosphorylated by Doc [55], which prevents it from dimerizing with its proapoptotic companion Bax, resulting in apoptosis. However, in the present study, only the IC_50_ levels for Doc in the PC-3 cell lines are in line with the IC_50_ observed by Yang et al. [56], but not in the DU-145 cells [56]. In fact, in our study, the observed IC_50_ level for Doc in the DU-145 cells is half of what other studies have reported. This discrepancy is easily addressed when we look at the different methods for cell viability quantification that have been employed in those studies, and it is well-known that the different methods used have different molecular targets and different mechanisms of action. Further, the results of this study with monotherapy with Que and the IC_50_ levels for Que obtained suggest that the mechanisms of action of Doc and Que are distinctively independent because the doses for IC_50_ required are distinct. The mechanism of action of Que reported by us in a previous study is via the downregulation of HSP-90, which leads to a sequence of downstream events that eventually result in apoptosis. However, other studies have suggested that the inhibitory effect of Que on cancer cell growth may be attributed to the inhibition of survival signaling proteins, such as protein kinase C (PKC-α) and the activation of death signals, such as PKC-δ [57]. Furthermore, Que is also reported to induce pro-apoptotic effects via mechanisms involving antioxidant effects and by the suppression of the p53 gene and BCL-2 protein [58]. In addition, the observed IC_50_ levels with monotherapy of Que conform to the results of most other reported studies and our own report earlier on IC_50_ levels of Que in CaP cells [22]. However, the results of this study do not confirm the time-dependent nature of Que’s effect, as reported by others [59]. In the present study, we observed overlapping IC_50_ values at 24 and 48 h and did not observe a decrease in the IC_50_ levels for Que at 48 h, as described by Zhaorigetu et al. [59]. This observation, however, contradicts earlier observations, and can easily be explained by the initial number of cells that we started the assays with. It is well known that if we start with a higher number of cells, cell death due to overcrowding occurs at lower IC_50_ levels [60]. 

The anticancer agents need to be present at the right concentration for them to interact with their target and have a pharmacodynamic impact. Cellular metabolism, interactions with transporters, and interactions with simultaneously administered anticancer drugs eventually affect the pharmacokinetic profile of a cytotoxic drug combination. Our treatment approaches and combined doses of Doc + Que, which were studied by altering the drug concentration and mixing ratios, were guided by the general knowledge that we had on drug interactions. We evaluated the combination index (CI) values for the DOC/Que combination (Figure 2), and identified synergism, additivity, and antagonism. We found that in both DU-145 and PC-3 cells, the TAP-1 approach, mainly at a low and mid concentration of Doc, could achieve either synergy or additive effects. However, when we used a higher concentration of Doc (2–5 nM), it resulted only in antagonism (TAP:1; Figure 2A). In contrast, pre-treatment with Que for 24 h followed by Doc for another 24 h resulted either in synergy or additive effects at all concentrations of both drugs (Figure 2A,B). When we used TAP-3 as the treatment approach (TAP:3; Figure 2A,B bottom panel), we observed only antagonism in both cell lines as the main effect. Upon review of the literature, we did not come across any study of this nature with an extensive Doc and Que combination and different TAPs employed to comparatively analyze and discuss the results of our study. The closest study that we could identify as a comparable study with Que and Doc combinations is reported in breast cancer cells [61]. However even in this study, a much higher single concentration of Doc was used, and the only TAP applied was simultaneous treatment of both drugs. Despite these differences, this is the closest study we could use for comparative analysis, and the high dose of Doc used is justifiable because the IC_50_ values of Doc breast cancer cells are generally higher [62]. The authors of the breast cancer study conclude that the concomitant use of Doc and Que led to cell growth inhibition associated with the induction of apoptosis and inhibition of cell survival, which is in line with the results the present study for TAP-2. The results of our study can also be compared with two other studies by the same group, where they show that the combination of Que and green tea sensitizes CaP cells to Doc treatment in an in vivo model of CaP [24,63]. The authors of these studies observed that the combination of green tea and Que with Doc significantly enhanced the potency of Doc twofold and reduced tumor growth by 62% compared with Doc alone in a 7-week intervention study [63]. The authors also observed a decrease in Ki67 and an increase of cleaved caspase-7 in tumors treated by the mixture, along with lowered blood concentrations of growth factors such as VEGF and EGF [63]. The results of our study, although performed only in vitro, are broadly consistent with the results of this study. Our study showed that the combination was effective only using TAP-1 or TAP-2 and that TAP-3 was completely ineffective where the main observation was antagonism. The more likely explanation for the synergistic effect of TAP-1 at low concentrations of Que and low to mid concentrations of Doc is the antioxidant effect of Que [31]. However, when TAP-2 was employed, the most observed effects were synergy and additive. This is more likely due to the antioxidant and apoptosis-inducing effects of Que. We have earlier published that a major target of Que in CaP Cells is the downregulation of stress protein HSP-90 [22]. Therefore, it is quite possible that, when we employed TAP-2, apart from the antioxidant effect of Que, there was also an induction of apoptosis due to the downregulation of the HSP-90 protein, resulting in increased cell death leading to synergy or additive effects at most of the doses studied. Antagonism observed in TAP-3 approaches can be partially explained based on P-glycoprotein expression and drug interaction, as cancer cells express efflux pump P-glycoprotein to efflux the toxic drug out [64]. In the TAP-3 approach that we employed, it is more probable that with pre-treatment with higher doses of Doc, the expression of p-glycoprotein was at a much higher level, resulting in the efflux of Doc from the cell. When these cells were subsequently treated with Doc, and Que simultaneously, the cells could efflux the drugs from their system and were unable to produce the desired effect.

Despite Doc being the first-line therapy for CRPC [65], the emergence of Doc resistance is a major problem for the clinical success of this therapy [66]. An objective of our study was also to identify the dose and treatment approaches with Que that can not only sensitize the CaP cells but also reverse the resistance in Doc-R in CaP cells. We selected Que for the combination treatment approach with Doc since it is shown to have promising anti-cancer properties in CaP and other cancers by targeting multiple signal pathways [17,28,30]. Moreover, Que has also been reported as a cost-effective agent to reverse chemo-resistances in hepatocellular carcinoma and breast cancer [67,68,69]. In addition, studies by Wang et al. showed that combining green tea and Que could sensitize CRPC to Doc [63]. Therefore, we evaluated the effect of the most beneficial TAP observed in this study, TAP-2, on Doc/R CaP Cells that we established in our laboratory. In this study, two Doc-R CaP cell lines (DU-145/R and PC-3/R) were established by repeated treatment of surviving DU-145 and PC-3 to progressively increasing concentrations of Doc. The extent of resistance established in the DU-145/R and PC-3/R cells was validated by performing a proliferation assay. Our results confirm the establishment of Doc/R as the administration of a higher concentration of Doc did not have any measurable effect on cell death (Figure 3A,B), although Que when tested independently, did have a measurable smaller effect on cell death of both resistant cells. More significantly, the combination therapy of Doc and Que caused much more cytotoxicity than Que alone (Figure 3E,F). The combined therapy using the TAP-2 modality resulted in considerable inhibition of cell migration, invasion, colony formation, and reduced cell viability (Figure 4, Figure 5 and Figure 6). The same treatment approach resulted in increased apoptosis (Figure 7). Therefore, based on the results of this study, we suggest that pre-treatment with non-nephrotoxic doses of Que followed by therapy with Doc, rather than increasing Doc dosage, as a strategy to overcome Doc resistance. Furthermore, combining Que with Doc might be more cost-effective and cause fewer side effects than Doc alone.

Despite the interesting results, we acknowledge that there are some limitations in this study. The first limitation is that we did not carry out experiments to decipher the mechanism(s) underlying the reversal of Doc resistance with the TAP-2 approach. The mechanism proposed here and based on the literature review is entirely speculative. Based on the literature review, it is proposed that Que interferes with multiple pathways involved in cancer cell proliferation and survival (Figure 8). We believe that one reason for the synergetic effect observed in the TAP-2 approach used in our study with low doses of Que and Doc is the antioxidant activity of Que. At low doses, Que’s known role of scavenging ROS by upregulation of enzymes involved in scavenging, such as superoxide dismutase (SOD), glutathione peroxidase (GxP), and catalase, is effective. Therefore, it is conceivable that by pretreating CaP cells with Que, locally generated ROS are scavenged and subsequent treatment with Doc leads to enhanced cancer cell killing. Further, Que’s ability to increase glutathione (GSH) levels in the cell can also lead to an increase in ROS scavenging (Figure 8A [1]). Apart from this, there are other therapeutic pathways which can lead to Que’s synergy with Doc. An important pathway in this synergy is the direct role of Que in decreasing the level of P-glycoprotein (P-gp), a multidrug resistant protein involved in Doc expulsion from the cell. This action of Que leads to increased intracellular levels of Doc, resulting in enhanced killing of cancer cell resulting in synergy (Figure 8B [2]). Other pathways, such as Que’s ability to modulate of HSP-90 levels, could also play an important role. Our earlier study showed that Que can destabilize HSP-90, resulting in apoptosis of cancer cells. Therefore, it is possible that the observed synergy obtained by pre-treatment with Que triggers apoptosis through this mechanism as well (Figure 8B [3]), Another possible reason for the synergistic action on pretreatment with Que is the effect of Que on androgen receptor (AR). Que is known to downregulate AR transcription levels, which subsequentially leads to decreased proliferation and increased apoptosis (Figure 8B [4]). A second limitation of this study is that the in vitro doses and TAP have not been corroborated in vivo. The in vivo study is pending IACU approval and when performed, will be a large study beyond the scope of this article.

## 5. Conclusions

In this study, we have demonstrated two important findings. The first is that pre-treatment for 24 h with Que sensitizes DU-145 (moderately aggressive) and PC-3 (very aggressive) CaP cells to lower doses of Doc. Secondly, this treatment approach is also beneficial in reversing Doc resistance in both DU-145/R and PC-3/R CaP cells. Although there are previous reports of Que’s ability to sensitize and reverse Doc resistance in CaP cells, this is the first study to delineate the exact combination of both drugs to be used and the treatment approach to be followed. The results of this and similar studies could make Que a suitable candidate for combination therapy with Doc in prostate and other cancers by sensitizing cancer cells to chemotherapeutic and reverse the acquired resistance. 

## Figures and Tables

**Figure 1 cancers-15-00902-f001:**
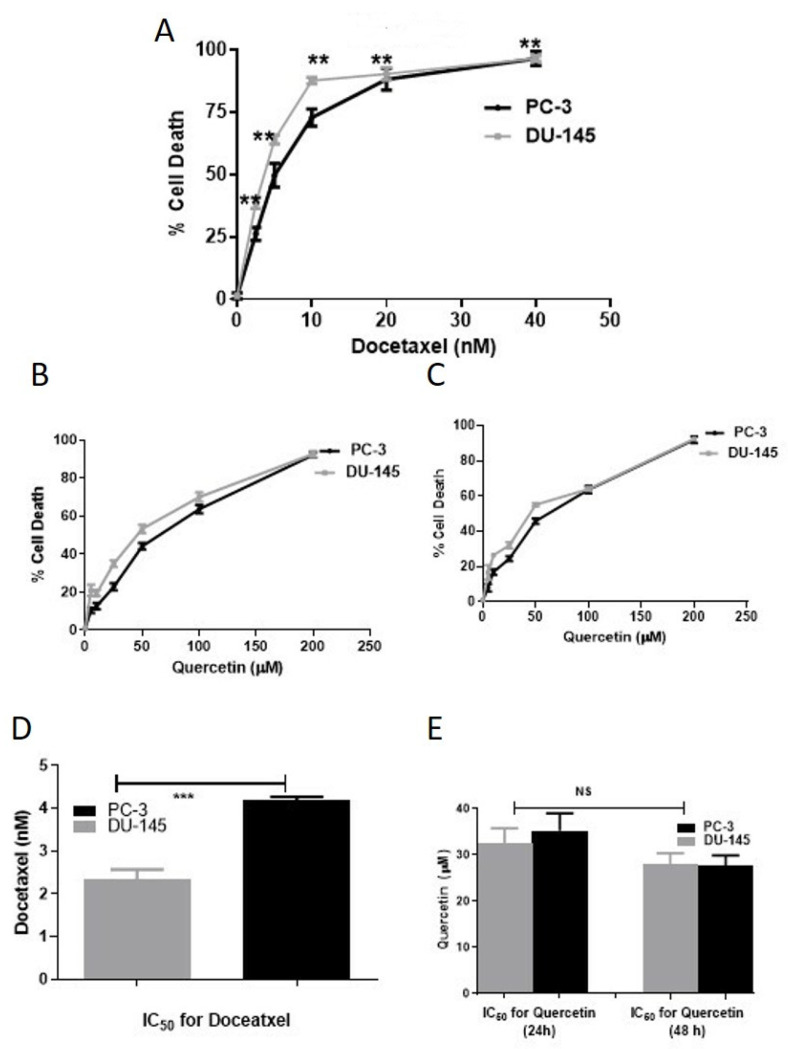
Cytotoxicity of docetaxel and quercetin on DU-145 (moderately aggressive) and PC-3 (very aggressive) human prostate cancer cells. The cells were treated with the indicated concentrations of docetaxel for 24 h and with quercetin for 24 and 48 h. (**A**) DU-145 and PC-3 cells treated with docetaxel (0.5–40 nM). ** *p* < 0.01. (**B**) DU-145 and PC-3 cells treated with quercetin (1.0–100 μM) for 24 h. (**C**) DU-145 and PC-3 cells treated with quercetin (1.0–100 μM) for 48 h. Cytotoxicity was evaluated by MTT assay. The data represent the mean ± SD of three experiments performed in triplicate. (**D**) A comparison of IC_50_ values obtained for DU-145 and PC-3 cells treated with docetaxel for 24 h. (**E**) A comparison of IC_50_ values obtained for DU-145 and PC-3 with quercetin for 48 h. ***: *t*-test *p*  <  0.001 between IC_50_ values obtained for the same drug in the same period and “NS” is not-significant.

**Figure 2 cancers-15-00902-f002:**
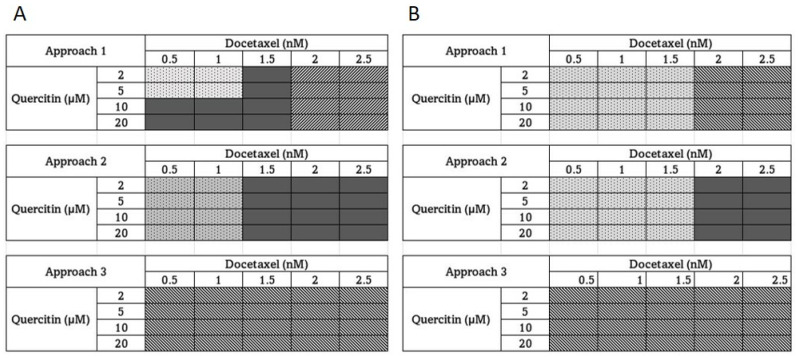
Combination cytotoxic effect of quercetin and docetaxel: (**A**) DU-145 (moderately aggressive), and (**B**) PC-3 (very aggressive human prostate cancer cells). Varying Que-Doc combinations were employed to treat the cells in accordance with three different treatment modalities: TAP:1 refers to simultaneous administration of both drugs for 24 h; TAP:2 to pre-treatment with Que for 24 h, then 24 h of treatment with Doc alone; and TAP:3 to pre-treatment with Doc for 24 h, then 24 h of simultaneous administration of both drugs. Cell viability (MTT assay) was assessed following the treatments and the combination index was calculated (CI). The CI values are presented as heat maps, with CI values < 0.9 (filled with dotted square) denoting a synergistic effect, CI values 0.9  ≤  CI  ≤  1.1 (filled with grey squares) denoting an additive effect, and CI value > 1.1 (filled with hatched squares) denoting an antagonistic effect.

**Figure 3 cancers-15-00902-f003:**
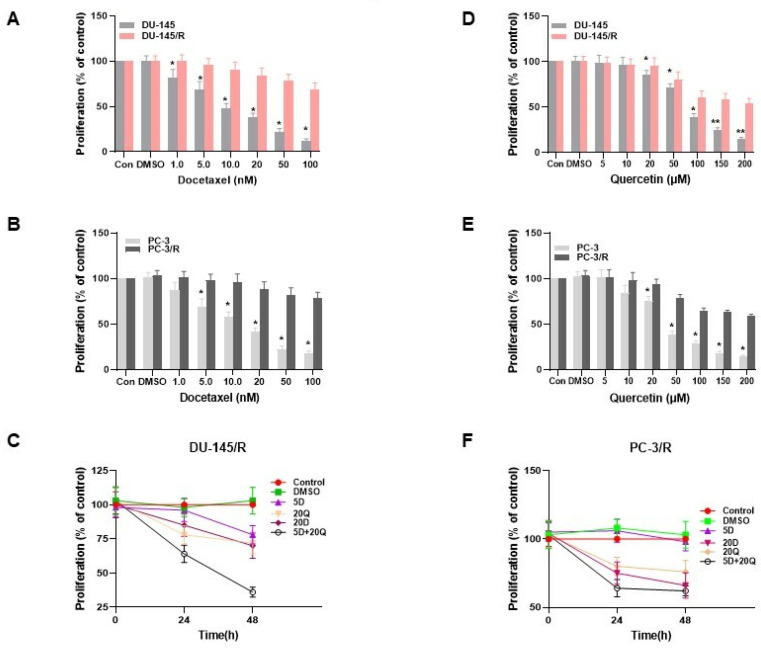
Generation of docetaxel-resistant prostate cancer cell lines and reversal of docetaxel resistance of quercetin using TAP-2 on proliferation. The generated docetaxel-resistant prostate cancer cells and docetaxel-naïve prostate cancer cells were treated with different concentrations of docetaxel or quercetin as indicated. 24 h later, the viability was measured using an assay as described in Section 2. (**A**) DU-145/R or DU-145 treated with docetaxel (nM). (**B**) PC-3-- or -PC-3/R treated with docetaxel (nM). (**C**) DU-145/R treated with different concentrations of docetaxel or quercetin or their combination using TAP-2 approach (1.0 nm of docetaxel and 20 µM of Quercetin) (**D**) DU-145 or DU-145/R treated with quercetin(µM) (**E**) PC-3 or PC-3/R treated with quercetin (μM). PC-3/R (**F**) PC-3/R treated with different concentrations of docetaxel or quercetin or their combination using the TAP-2 approach (1.0 nm of docetaxel and 20 μM of Quercetin as indicated). Cell viability was determined at 24 h or 48 h. Results are the mean ± SD of three independent experiments performed in triplicate. * *p* < 0.05, ** *p* < 0.01, compared with control group of both cell lines.

**Figure 4 cancers-15-00902-f004:**
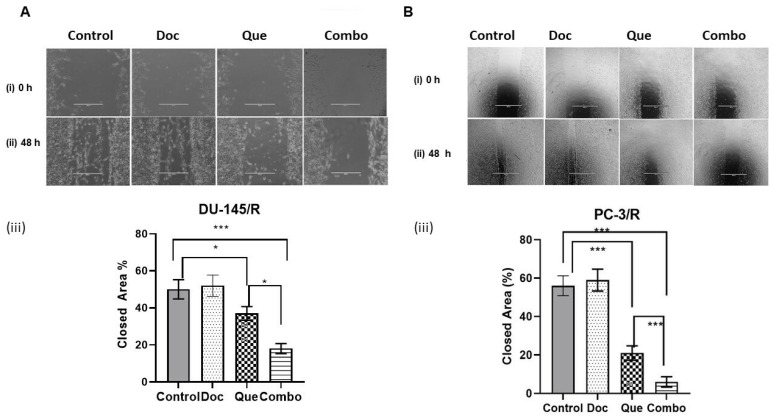
Effects of docetaxel and docetaxel + quercetin on migration of resistant prostate cancer cell lines DU-145/R and PC-/R. The cells were treated as described in Section 2 with either the drugs individually or in combination and were evaluated with respect to their effects on the migration after 48 h. (**A**) DU-145/R cells and (**B**) PC-3/R cells. We performed three experiments in triplicate and the data are presented as means ± S.D. * *p* < 0.05, *** *p* < 0.001. In both (**A**) and (**B**), (i) is a cell image at 0 h, (ii) is a cell image at 48 h, and (iii) shows a graphic representation of the Image J software quantification of % closed area compared with the control.

**Figure 5 cancers-15-00902-f005:**
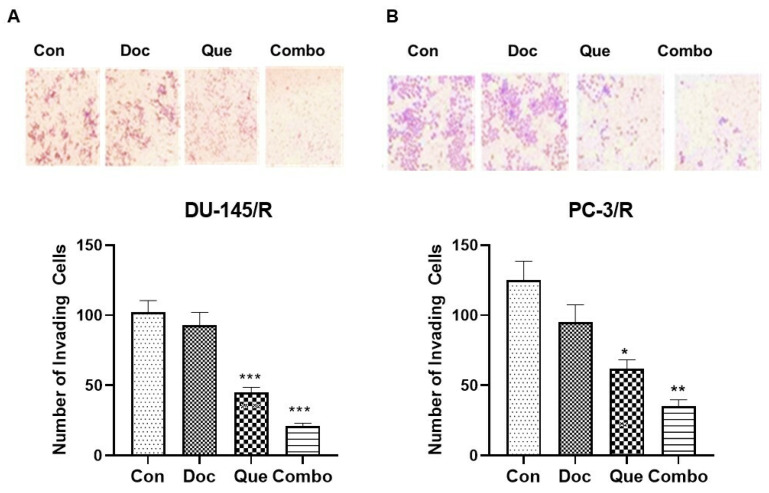
Invasive activity of docetaxel-resistant prostate cancer cell lines DU-145/R (**A**) and PC-3/ (**B**) in treatments with different drugs and drug combinations. Cells were applied to the upper surface of a filter coated with basement membrane proteins using a cell invasion assay, as described in Section 2. After incubation for 48 h, the upper surface of the filter was scrubbed free of cells and protein, the filter was fixed and stained, and the lower surface was photographed. The top panel shows representative photographs at a magnification of ×100: Lanes 1–4 are the control, docetaxel, quercetin, and combination treatments, respectively. The bottom panel in each figure shows quantitative values of the total number of cells invading through the filter. * *p* < 0.1, ** *p* < 0.01 *** *p* < 0.001.

**Figure 6 cancers-15-00902-f006:**
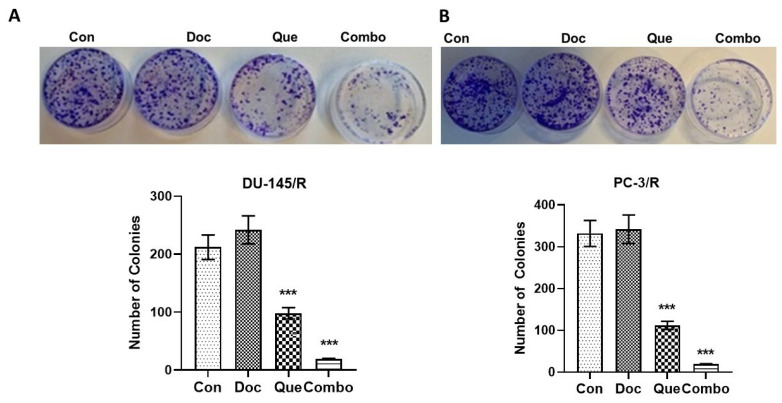
Effects of Doc and docetaxel + quercetin on colony-forming ability of resistant prostate cancer cell lines DU-145/R and PC-/R. The cells were treated as described in Section 2 with either the drugs individually or in combination and evaluated with respect to their effects on colony formation after 8 days. (**A**) DU-145/R cells and (**B**) PC-3/R cells. We performed the experiments in triplicate and the data are presented as means ± S.D. *** *p* < 0.001. The top panel shows representative images of the plate after 8 days. The bottom panel is the graphic representation of the mean and standard deviation of the number of colonies of three independent experiments performed in triplicate. The statistical significance of the difference between the control and treated cultures was calculated by Student’s *t*-test.

**Figure 7 cancers-15-00902-f007:**
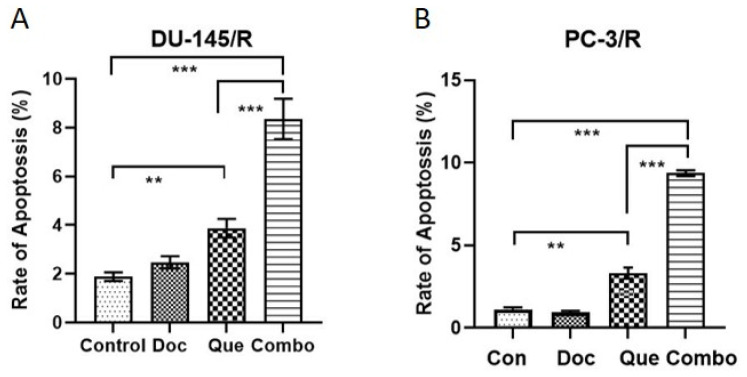
Induction of apoptosis by quercetin in docetaxel-resistant cells. The percentage (%) of cell distribution (**A**) of DU-145/R cells and the percentage (%) of cell distribution (**B**) of PC-3/R cells 24 h after treatment with designated drugs or vehicle control, as determined by Annexin V-7AAD and PE staining. Data are presented as mean ± SD. ** *p* < 0.01 *** *p* < 0.001. The experiments were performed at least three times in triplicate.

**Figure 8 cancers-15-00902-f008:**
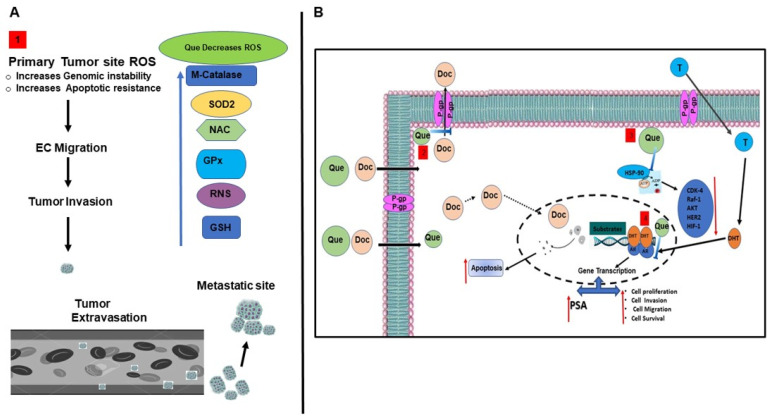
Schematic diagram showing the potential pathways by which quercetin synergistically enhances the docetaxel effect. (**A**). Antioxidant Effect: (1) Que increases ROS scavenging by increasing the levels of enzymes involved in this pathway, such as superoxide dismutase (SOD), glutathione peroxidase (GxP), and catalase, and non-enzymatically by increasing the level of Glutathione (GSH). (**B**) Therapeutic Effect: (2) Que decreases the level of P-gp, a multidrug resistant protein involved in Doc expulsion, thereby increasing the intracellular levels of the drug and resulting in synergy; (3) Que can destabilize HSP-90, which results in apoptosis by downregulating the levels of cyclin-dependent kinase 4 (CDK-4), rapidly accelerated fibrosarcoma (Raf-1) kinase, human epidermal growth factor receptor-2 (HER-2) and hypoxia induciblefactor-1 (HIF-1) of cancer cells, resulting in synergy with Doc action; and (4) Que can decrease the level AR which eventually decreases the levels of protein involved in cell proliferation and cell survival, thereby contributing to synergistic action with Doc. Abbreviations: Que, quercetin; Doc, docetaxel; T, testosterone; HSP-90, heat shock protein-90; T, testosterone; DHT, dihydroxy testosterone; and AR, androgen receptor. 
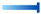
 indicates inhibition or blockade with induction, and the red arrow indicates up or down regulation based on direction.

## Data Availability

The data presented in this study are available on request from the corresponding author.

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
