# Peer review of "Combination Modality Using Quercetin to Enhance the Efficacy of Docetaxel in Prostate Cancer Cells"

_cancers, 2023, doi:10.3390/cancers15030902_

Round 1
Reviewer 1 Report
1) The quality of the figures needs to be improved. Very bright colors are used that make it difficult for the reader to concentrate.
2) The mechanism of action of these compounds should be presented in the scheme.
3) Mechanisms of therapy of cancerous tumors with the help of substances with antioxidant properties should be clearly indicated in the introduction. Due to the impact on which signaling and metabolic pathways, antioxidants are able to suppress tumors.
4) the conclusion should be shortened
5) The abstract should be written in more detail so that it reflects in more detail the problems, the prerequisites for starting these studies and the results clearly obtained by the authors.
6) The effects of taxifolin and nanoparticles carrying taxifolin on normal cells should be discussed. https://pubmed.ncbi.nlm.nih.gov/30776416/ https://pubmed.ncbi.nlm.nih.gov/36432668/
Author Response
Reviewer 1 comment 1: The quality of the figures needs to be improved. Very bright colors are used that make it difficult for the reader to concentrate
Response: We thank the reviewer for pointing this. The entire set of figures are now modified using only standard black and white graphical illustration with different fill patterns. The revised figures have been uploaded in place of the old figures in the main document.
Reviewer 1 comment 2: The mechanism of action of these compounds should be presented in the scheme.
Response: We appreciate the reviewers comment and recognize the justification for a summary figure for the mechanism of action. A new Figure (Figure 8) summarizing the mechanism of action of the compounds is added. This has led to the discussion on this section on page 15 line 591-616
Reviewer 1 comment 3: Mechanisms of therapy of cancerous tumors with the help of substances with antioxidant properties should be clearly indicated in the introduction. Due to the impact on which signaling and metabolic pathways, antioxidants are able to suppress tumors.
Response: In the introduction section this information is added on page 2 lines 87-93 and line 98.Aslo, information suggesting the chemopreventive action of antioxidant drug Quercetin has been added on page 3 lines 101-102, 105-106, 112-113 and 116-118.
Reviewer 1 comment 4: The conclusion should be shortened.
Response: The conclusion has been substantially shortened. The original conclusion section of was 20 lines and now it is reduced by more than 55% to 9 lines. Also, the reduction required rewording of line 621, 623,624,628 and 629. The changes have been made using the track change options using word processor.
Reviewer 1 comment 5: The abstract should be written in more detail so that it reflects in more detail the problems, the prerequisites for starting these studies and the results clearly obtained by the authors.
Response: We agree with the reviewers and this section as indicated earlier has been completely rewritten. This change has been incorporated in the text and is introduced as highlighted text.
Reviewer 1 comment 6: The effects of taxifolin and nanoparticles carrying taxifolin on normal cells should be discussed.
Response: The summary of the manuscript suggested by the reviewer is now added to the text in the discussion section on page 12 line 455- page 13 line 460. Two new references number 45 and 46 have been cited. As a consequence of this the total number of references have increased to 69 from in the original version.
Reviewer 2 Report
1) The abstract should be written in more detail.
2) After which passage the cells were used for experiments. Whether and by what method analyzes of cell culture for contamination with mycoplasma were carried out.
3) 2.5. Clonogenic assays: Remove bold type
4) The quality of the images in Figure 4 should be significantly improved.
5) However, the molecular mechanism of the synergistic action of the studied agents has not been shown. Synergistic effects are demonstrated, while much attention is paid to molecular mechanisms in the discussion. The discussion is largely speculative. In the discussion, the authors are invited to provide a signal scheme (including based on the analysis of the literature) of the action of the investigated agents and clearly indicate the contribution of the presented experiments to the understanding of the mechanisms. Or the discussion should be substantially reworked.
Author Response
Reviewer 2 comment 1: The abstract should be written in more detail
Response: The abstract is completely rewritten as indicated in response to reviewer 1.
Reviewer 2 comment 2: After which passage the cells were used for experiments. Whether and by what method analyzes of cell culture for contamination with mycoplasma were carried out.
Response: The details sought by the reviewer are now included in the methods section on page 2 lines 143-144 and line page 4 lines 151-153 as highlighted text.
Reviewer 2 comment 3: 2.5. Clonogenic assays: Remove bold type
Response: Bold type has been removed.
Reviewer 2 comment 4: The quality of the images in Figure 4 should be significantly improved.
Response: The quality of Figure 4 has been improved with new images and the graphical illustration has been changed from color to black and white with different fill patterns.
Reviewer 2 comment 5: However, the molecular mechanism of the synergistic action of the studied agents has not been shown. Synergistic effects are demonstrated, while much attention is paid to molecular mechanisms in the discussion. The discussion is largely speculative. In the discussion, the authors are invited to provide a signal scheme (including based on the analysis of the literature) of the action of the investigated agents and clearly indicate the contribution of the presented experiments to the understanding of the mechanisms. Or the discussion should be substantially reworked.
Response: A new figure, figure no 8 has been included to reflect the comment suggested by the reviewer and the figure has been discussed on page 15 line 591-615.
Reviewer 3 Report
In this article, the authors evaluated the dose ranges at which the natural flavonoid quercetin synergies and sensitizes prostate cancer cells to docetaxel therapy and reversal of docetaxel resistance.
The topic is interesting, the paper is well-written and organized. However, there is still some room for improvement and I have the following comments for that.
A summarized scheme with all steps of the study would be welcome to attract readers.
Introduction: the authors must make a clearer difference between the chemopreventive and chemotherapeutic roles of Quercetin along with its mechanisms of action based on the results of scientific studies.
What perspectives for human health does this MS have?
Lines 552-554: Mention more limitations of this study not only one.
Consider revision accordingly.
Author Response
Reviewer 3 comment 1: A summarized scheme with all steps of the study would be welcome to attract readers.
Response: A new figure, figure no 8 has been included to reflect the summary of actions and discussed on page 15 line 591-615.
Reviewer 3 comment 2: Introduction: the authors must make a clearer difference between the chemopreventive and chemotherapeutic roles of Quercetin along with its mechanisms of action based on the results of scientific studies.
Response: The changes suggested by the reviewer is introduced in the introduction section on page 2 line 98 and on page 3 lines 101, 105, 112 and line 116.
Round 2
Reviewer 1 Report
All my comments have been taken into account. The article can be accepted for publication in its current form
Reviewer 2 Report
my comments are taken into account by the authors. article can be accepted for publication
Reviewer 3 Report
I endorse publication.